# Transform Trained Transformer for Accelerating Native 4K Video Generation

**Jiangning Zhang** [1 2]  **Junwei Zhu** [2]  **Teng Hu** [2]  **Yabiao Wang** [1 2]  **Donghao Luo** [2]  **Weijian Cao** [2]  **Zhenye Gan** [2]
**Xiaobin Hu** [3]  **Zhucun Xue** [1]  **Xiangtai Li** [4]  **Chengjie Wang** [2]  **Yong Liu** [1]

## Abstract

Native 4K (2176×3840) video generation remains a critical challenge due to the quadratic computational explosion of full-attention as spatiotemporal resolution increases, making it difficult for models to strike a balance between efficiency and quality. This paper proposes a novel Transformer retrofit strategy termed T3 (**T**ransform **T**rained **T**ransformer) that, without altering the core architecture of full-attention pretrained models, significantly reduces compute requirements by optimizing their forward logic. Specifically, T3-Video introduces a multi-scale weight-sharing window attention mechanism and, via hierarchical blocking together with an axis-preserving full-attention design, can effect an "attention pattern" transformation of a pretrained model using only modest compute and data. Results on 4K-VBench show that T3-Video substantially outperforms existing approaches: while delivering performance improvements (+4.29↑ VQA and +0.08↑ VTC), it accelerates native 4K video generation by more than 10×. Demo and source code are available in #Supp.

## 1. Introduction

Media applications have an increasingly urgent demand for Ultra-High Definition (UHD) video generation. 4K (2160×3840) video, with its fine texture details and immersive visual experience, has become a core requirement in film production, virtual reality, advertising, and related fields. However, native 4K video generation (*i.e.*, end-to-end synthesis without relying on super-resolution or other post-processing steps) remains difficult for most models; the central bottleneck is the excessive computational cost that

originates from full-attention's "quadratic computational explosion" in transformers.

Studying native UHD generation is important for end-to-end systems and for guaranteeing performance, and is increasingly becoming a trend (Xue et al., 2025; Hu et al., 2026), but exploring 4K video generation exhibits clear shortcomings, which can be summarized into three core challenges: ***First**, the inherent trade-off between computational efficiency and quality.* Most methods adopt a "low-resolution generation + video super-resolution" cascaded pipeline (Blattmann et al., 2023b; Wang et al., 2025c; Zhang et al., 2025i), but the high-frequency details added in super-resolution often lack semantic consistency and cannot deliver true 4K quality. Few attempts at native 4K generation (*e.g.*, UltraWan (Xue et al., 2025)) rely on full-attention architectures and demand huge compute. As shown in Fig. 1, directly running inference with pretrained Wan-T2V-1.3 (Wang et al., 2025a) and HunyuanVideo (Kong et al., 2024) yields unsatisfactory results and is extremely slow. ***Second**, waste of pretraining resources.* Current video foundation models (*e.g.*, Sora (OpenAI, 2024), HunyuanVideo (Kong et al., 2024), Wan series (Wang et al., 2025a)) require massive data and computation to pretrain, but existing efficient techniques (*e.g.*, VSA (Zhang et al., 2025g) and FPSAttention (Liu et al., 2025a)) typically modify model architectures or weights substantially, preventing reuse of pretrained weights and forcing full retraining. ***Third**, insufficient architectural compatibility and generalization.* Only a handful of leading companies can afford pretraining resources for new architecture designs. Some methods introduce dedicated modules (*e.g.*, LinGen (Wang et al., 2025b)) to achieve linear complexity, but such designs are poorly compatible with the mainstream Transformer ecosystem. Recently UltraGen (Hu et al., 2026) explored native 4K video generation for 29 frames, requiring training on over a hundred GPUs, but fails to produce semantically rich scene generation, as shown in Fig. 1.

To break through the above bottleneck, we propose T3-Video, a one-line code replacement for naive "full-attention transformation" that achieves linear computation scaling by optimizing attention logic without altering the Transformer architecture. We reconstruct conventional global single-scale attention into multi-scale shared-window atten-

[1]APRIL Lab, Zhejiang University [2]Youtu Lab, Tencent [3]National University of Singapore [4]Peking University. Correspondence to: Chengjie Wang <jasoncjwang@tencent.com>, Yong Liu <yongliu@iipc.zju.edu.cn>.

*Proceedings of the 43rd International Conference on Machine Learning*, Seoul, South Korea. PMLR 306, 2026. Copyright 2026 by the author(s).

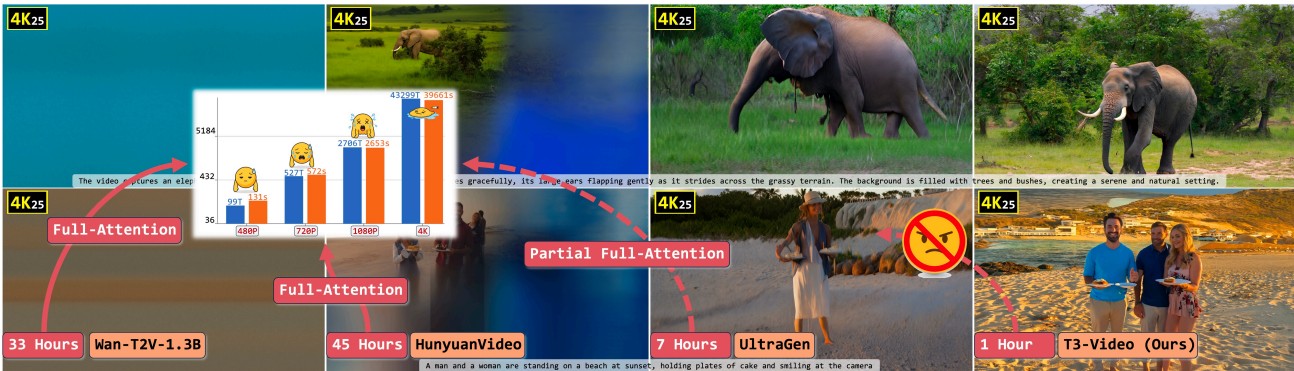

*Figure 1.* 4K (2176×3840) inference visualization: Wan2.1-T2V-1.3B (Wang et al., 2025a) (81*f*), HunyuanVideo (Kong et al., 2024) (41*f*), UltraGen (Hu et al., 2026) (29*f*), and our T3-Video-T2V-1.3B (81*f*). Efficiency tests are performed on 81 frames with FlashAttention2 (Dao, 2023) on a single H20 GPU. Bar chart: blue denotes theoretical MACs while orange denotes measured latency of DiT in Wan2.1-1.3B (Wang et al., 2025a). Vertical axis is on a logarithmic scale.

tion: tokens are partitioned into non-overlapping windows at multiple scales with shared parameters for cross-scale information exchange. This preserves full-attention's global semantic modeling while reducing complexity from $\mathcal{O}(L^2)$ to $\mathcal{O}(L \times L_b)$ ($L$: total tokens, $L_b$: tokens per window). The method requires no core modification and enables one-line code replacement of existing attention layers, maximizing pretrained resource reuse for efficient UHD-4K generation fine-tuning. In summary, our contributions are threefold:

- An elegant T3 attention optimization paradigm: reconstruct global full attention as multi-scale shared window attention, combined with hierarchical blocking and axis-preserving strategies. While remaining compatible with pretrained weights, this reduces computational complexity from $\mathcal{O}(L^2)$ to $\mathcal{O}(L \times L_b)$, solving the computational explosion in 4K video generation without modifying the Transformer core architecture.

- An efficient 4K video generation and deployment framework: based on T3 we design and generalize a series of Wan-based 4K/81*f* generation video models, integrating LoRA fine-tuning, Step/CFG distillation, and a lightweight eVAE for extreme efficiency optimization, achieving native 4K 81-frame inference with memory <60G and one-hour runtime on a single H20 GPU.

- Extensive experiments on 4K-VBench validate the effectiveness and generality of the method, *e.g.*, T3-Video-T2V-1.3B versus UltraGen yields +4.29 VQA, +0.08 VTC, 7× speedup in 4K inference, achieves the first native 81-frame 4K generation, and shows strong performance on I2V/T2V tasks with 1.3B/5B models.

## 2. T3-Video

### 2.1. Why Full-Attention for Diffusion Video?

A survey of recent video diffusion models shows that they largely retain Transformer-based architectures (Xing et al., 2024), with few attempts to explore novel structural modifications; the main reasons are as follows:

**Inclined DiT paradigm.** Since the emergence of Sora sparked video generation based on DiT (Peebles & Xie, 2023), its Transformer-based structure has been defaultly adopted by subsequent video foundation models (Hong et al., 2023; Kong et al., 2024; Wang et al., 2025a) due to its excellent and stable performance, as well as its simple and easy-to-implement code.

**Hardware support and community optimization.** Since the ViT era (Dosovitskiy et al., 2021), the community has developed many effective methods to accelerate ViT training and inference. Examples include recent works such as TeaCache (Liu et al., 2025b), KV-Cache (Pope et al., 2023), and the FlashAttention (Dao et al., 2022; Dao, 2023; Shah et al., 2024)/SageAttention (Zhang et al., 2025e;b;d) families. These approaches target structure-agnostic inference acceleration for naive full-attention to enable better practical deployment, whereas linearized attention schemes (Choromanski et al., 2021; Katharopoulos et al., 2020) are harder to adapt and still suffer from subpar performance and stability.

**High-cost trial and error leads to the absence of new structures.** Pretraining video foundation models requires massive private data, compute, and time, feasible only for few companies. Unlike the vision-backbone era's diverse architectures, experimenting with novel designs is impractical due to high trial-and-error costs and long iteration cycles. Researchers typically focus on downstream applications, fine-tuning pretrained models with modest resources without architectural changes. This stems from full self-

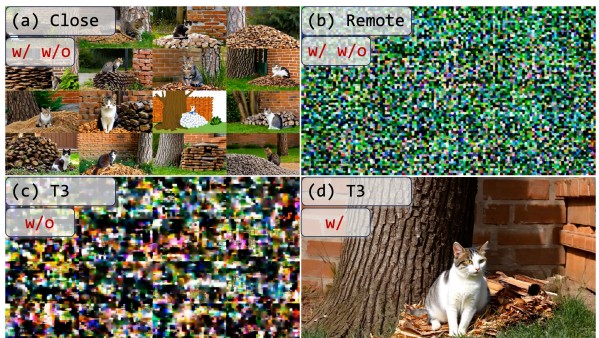

*Figure 2.* 720P results w/ or w/o finetuning for close/remote window-attention and T3 module by 4×4 blocks.

attention's quadratic compute growth with increasing tokens, causing a "long-sequence computational explosion." This is critical given social-media-driven demand for high-resolution applications: doubling resolution quadruples tokens, and attention's quadratic computation creates staggering workloads at UHD-4K, as shown in Fig. 1. While some works attempted training linear-complexity video models from scratch (Wang et al., 2025b) or introducing autoregressive frameworks (Wang et al., 2024), these haven't gained traction due to performance and efficiency limitations. Full-attention remains dominant in current video models, raising the question: *can we preserve attention-based pretrained structures while achieving approximately linear MACs?*

### 2.2. Transform Trained Transformer (T3)

This paper studies the computational explosion caused by long token sequences in high-resolution video generation, and aims to use minimal architectural adaptations to reduce compute demands while preserving and leveraging pretrained weights so that models can be fine-tuned with only modest compute and data.

**Ideal optimized objective of full-attention.** For a pre-trained DiT-based video foundation model, we believe an ideal acceleration solution should satisfy: 1) maximally inherit pretrained weights to minimize the cost of re-pretraining; 2) change the self-attention computation as little as possible (or not at all), so that community acceleration components can be leveraged and instability is reduced; and 3) require only modest additional compute to restore the model's video-generation quality while delivering a stable and substantial acceleration factor.

**Window-attention as the core solution.** Taking batch size 1 as an example, the input video latent features are represented by a tensor $F \in \mathbb{R}^{C \times T \times H \times W}$, where $C$ denotes the number of channels and $T, H, W$ denote the number of grid locations along the temporal, height, and width dimensions, respectively. The total number of tokens is $L = THW$. For a standard Transformer layer, its number of parameters and Multiply-Accumulate operations (MACs), ignoring bias

terms for analytical convenience, are given by:

$$\text{Param.}_{full} = \underbrace{4C^2}_{Param._{proj.}} + \underbrace{0}_{Param._{attn.}} + \underbrace{2CC_{ffn}}_{Param._{ffn}}.$$

$$\text{MACs}_{full} = \underbrace{4LC^2}_{MACs_{proj.}} + \underbrace{2L^2C}_{MACs_{attn.}} + \underbrace{2LCC_{ffn}}_{MACs_{ffn}}.$$

For windowed attention partitioned into $N_b = n_t \times n_h \times n_w$ blocks, with each block containing $L_b = L/(n_t \times n_h \times n_w)$ tokens, the number of parameters remains unchanged, while the MACs for the attention component are significantly reduced as $L_b (\ll L)$ decreases:

$$\text{MACs}_{window} = \underbrace{4LC^2}_{MACs_{proj.}} + \underbrace{2LL_bC}_{MACs_{attn.}} + \underbrace{2LCC_{ffn}}_{MACs_{ffn}}.$$

Considering $L_b \ll L$ and $C \ll L$ for video generation, the bulk of attention MACs can be reduced by a factor of $n_t \times n_h \times n_w$. Moreover, as long as the base window size is fixed, MACs grow linearly with the number of tokens as video resolution increases. However, window attention is limited to local modeling and ignores global information within a single layer, which leads to poor global semantic consistency, as shown in Fig. 2-(a); even after fine-tuning it still cannot produce globally consistent videos.

**Shared window-attention as a concurrent global transceiver.** Inspired by EMOv2 (Zhang et al., 2025a), which introduces a novel i²RMB block to simultaneously model local and global information, we design window-attention as a simultaneous transceiver that enables bidirectional information exchange between local and global feature maps. In particular, this module inherits the standard full-attention module and only changes the computation mode without modifying the Transformer architecture itself.

- **Notation.** Given an input video latent feature (batch dimension omitted) $F \in \mathbb{R}^{C \times T \times H \times W}$, we use the subscript $(t, h, w)$ to denote a spatiotemporal voxel index ($t \in \{1, \ldots, T\}$, $h \in \{1, \ldots, H\}$, $w \in \{1, \ldots, W\}$). For a local spatiotemporal block we denote its tensor by $X \in \mathbb{R}^{C \times m_t \times m_h \times m_w}$, where $(m_t, m_h, m_w)$ are the window sizes along time, vertical and horizontal dimensions. Let $\text{ATTN}(\cdot)$ denote the standard attention operation, which acts on an input block of shape $C \times m_t \times m_h \times m_w$ and returns an output of the same shape.

- **Design of multi-scale discrete windows.** We replace the global attention operation by $S$ parallel local attention operations at multiple scales, all using the same fixed window size $(m_t, m_h, m_w)$. At each scale $s \in \{1, \ldots, S\}$, $F$ is partitioned into $n_t^{(s)} \times n_h^{(s)} \times n_w^{(s)}$ blocks with stride $(\Delta t_s, \Delta h_s, \Delta w_s)$. The strides are chosen so that the blocks form a disjoint tiling of the corresponding dimensions, *i.e.*, every voxel participates in exactly one window-attention computation at each scale.

- **Boundary scales.** The finest local scale is voxel-adjacent, *i.e.* $\Delta t_1 = 1, \Delta h_1 = 1, \Delta w_1 = 1$. The coarsest (remote) scale $S$ uses strides that evenly cover the entire domain: $m_t \Delta t_S = T$, $m_h \Delta h_S = H$, $m_w \Delta w_S = W$. Thus each block at scale $S$ uniformly covers the spatiotemporal grid, which is equivalent to the maximum-scale window spanning the whole video.

- **Computation and parameter sharing for local attention.** For any block $B_{i,j,k}^{(s)}$ at scale $s$, extract the corresponding input subtensor $X_{i,j,k}^{(s)}$ and apply a unified attention operation whose parameters are shared across all scales and all blocks: $Y_{i,j,k}^{(s)} = \text{ATTN}\left(X_{i,j,k}^{(s)}\right) \in \mathbb{R}^{C \times m_t \times m_h \times m_w}$. The meaning of parameter sharing is that for all blocks indexed by $s, i, j, k$, $ATTN$ uses the same set of projection matrices $W_Q, W_K, W_V, W_O$, so that the model parameter count remains unchanged while introducing a consistent attention pattern across different scales, which essentially also adds an inductive bias to the model and reduces the learning difficulty.

- **Aggregation of block outputs to the whole map.** Define for a given position $(t, h, w)$ the set of all blocks that contain that position as $\Omega(t, h, w) = \{(s, i, j, k) \mid (t, h, w) \in \text{support}(B_{i,j,k}^{(s)})\}$. We adopt a scale-weighted, normalized linear aggregation strategy. $\hat{F}[:, t, h, w] = \frac{1}{Z(t,h,w)} \sum_{(s,i,j,k) \in \Omega(t,h,w)} \omega_s \ Y_{i,j,k}^{(s)}$, Here $\omega_s \geq 0$ denotes the weight for scale $s$ (in this paper we default to $1/S$ since a learnable scalar did not yield further improvements), and the normalization factor is $Z(t, h, w) = \sum_{(s,i,j,k) \in \Omega(t,h,w)} \omega_s$, to ensure that the output at each position is a weighted average. Fig. 3 provides a schematic illustration using the 2D position (1,1) as an example.

**MACs-restricted hierarchical strategy.** The naive blocking strategy applies blocking along the $T, H, W$ axes and uses the same scheme for all layers. Empirically, however, this causes blocky spatial discontinuities and temporal jumps in the generated videos; although longer training can mitigate these artifacts, that clearly departs from the intent of the study. To address this, we propose a hierarchical strategy that, while keeping the $MACs$ within the intended budget, uses different blocking schemes for different layers and leverages overlaps between adjacent blocks to substantially improve transition smoothness. We group every 5 layers and cycle this configuration through the full depth.

**Axis-preserving full-attention.** Additionally, we propose an axis-preserving strategy that applies $n_t = 1$ or $n_h/n_w = 1$ to selected layers within each group to realize full-attention along the corresponding axis, improving generation stability while substantially reducing computational cost. Notably, this strategy and the aforementioned hierarchical strategy can be flexibly configured and switched according to constraints.

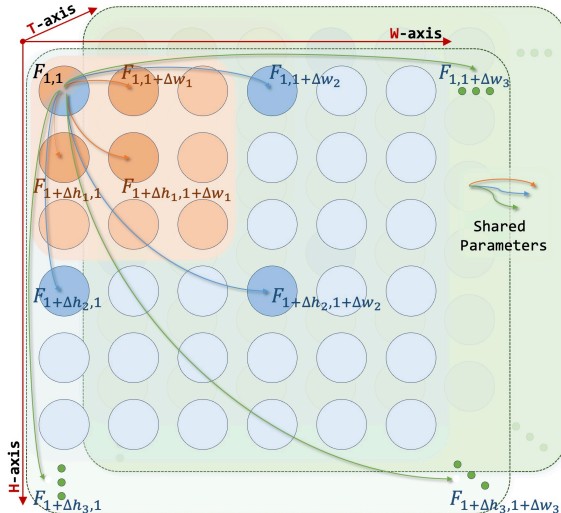

*Figure 3.* **Intuitive diagram for T3 strategy.** Taking the typical 2D (1,1) position $F_{1,1}$ of the input latent feature with a window size of 2 as an intuitive example, it uses shared attention parameters to perform information exchange across multiple scales ($\Delta h_1/\Delta w_1, \Delta h_2/\Delta w_2, \cdots$) simultaneously.

---

**Algorithm 1** T3-Video Module Pseudocode

---

```
class SelfAttention_T3_Video(SelfAttention):
    def __init__(self, dim, num_heads, eps,
    layer_idx, C, H, W, S):
        super().__init__(dim, num_heads, eps)
        self.n_thw = block_params(layer_idx, C, H, W)
        self.S = S

    def forward(self, x):
        q, k, v = self.qkv(x)
        xs = []
        for s in range(self.S)
            q_s, k_s, v_s = reshape(q,k,v,self.n_thw,s)
            x_c = ATTN(q_s, k_s, v_s)
            xs.append(recover(x_c, self.n_thw, s))
        x = torch.stack(xs).mean(dim=0)
        return self.o(x)
```

---

**Notes**: Newly added revisions are in **Green**. **block_params** means block configure calculated for each DiT layer.

**Implemented in one line of code.** Algorithm 1 shows the pseudocode of our transformed T3 attention module, which can be used as a drop-in replacement for the standard SelfAttention module to achieve code-agnostic speedups. To trade off effectiveness and efficiency in practical scenarios, we set the default scale $S$ to 2 in this paper. As shown in Fig. 2, the model fails to learn when using only the close (a) or remote (b) mode. Forcing the window attention to attend to both local and global contexts at once still yields outputs that indicate learning difficulty without training (c); therefore, we found that full fine-tuning easily restores video generation quality (detailed results in Sec. 3.2.1).

**Futile structure-preserving attempts.** We also explored several other small, structure-preserving modifications when designing the T3 module, such as: *1)* zero convolution for better modeling of local inductive bias; *2)* reducing the dimensionality of $K/V$ to further lower per-window computation; *3)* using an extra RoPE to strengthen positional

information within windows; *4)* employing a dynamic $\alpha$ to modulate the weights of windows at different distances; and *5)* replacing the parallel strategy with a cascaded one, *i.e.*, first processing at the fine scale, then processing the fused representations at the coarse scale. None of these produced noticeable positive improvements.

**Discussion with recent UltraWan and UltraGen.** *i)* **Methodologically**, UltraWan (Xue et al., 2025) was the first to fine-tune the original model at 4K (29 frames) without any acceleration, while UltraGen (Hu et al., 2026) adds complex, carefully designed modules for intermediate layers (keeping the original full-attention operation for the first and last two layers). T3-video, by contrast, only applies a small, more elegant tweak to the attention computation logic. *ii)* **In terms of training resources**, both two methods rely on 128 H20 GPUs for training, whereas we use only 64 GPUs. *iii)* **In terms of results**, our approach is more efficient: we are the first to enable native 4K resolution training and inference with 81 frames, and we obtain substantially better performance (+4.29 VQA and +0.08 VTC over UltraGen in Tab. 5) and efficiency ($7\times$ over UltraGen).

**Discussion with Shifted Windows, Neighborhood Attention, and i²RMB.** *i)* Compared with the shifted windows in SwinTransformer (Liu et al., 2021) and Neighborhood Attention in (Hassani et al., 2023), the T3-Video can model global information within a single layer via parallel close and remote paths, yielding both higher efficiency and better results. Typically, we replaced T3-module with shifted windows and trained it at 720P. Under the same 5K iterations, it only achieved 67.34 VQA/0.87 VTC, which is significantly lower than our 69.37 VQA/0.90 VTC (see Tab. 7). *ii)* Compared with the i²RMB block in (Zhang et al., 2025a), T3-Video only applies a structural transformation to attention (it does not create a new block nor change the core Transformer architecture), so it is more compatible and achieves better performance. This compatibility allows us to leverage pretrained weights to for the first time explore 4K 81-frame video generation, and it can also improve full-attention capability (see Sec. 2.6).

### 2.3. Why T3 Works?

**Full-attention and linear layers are a special case of T3.** When $(m_t, m_h, m_w)$ take their minimum values $(1, 1, 1)$, T3 degenerates to a linear layer with projection matrix $W_V$ (the attention matrix is $(1)$). When $(m_t, m_h, m_w)$ take their maximum values $(T, H, W)$, T3 instantiates full-attention. Both modules have been extensively validated (*e.g.*, MLP-Mixer (Tolstikhin et al., 2021) and ViT (Dosovitskiy et al., 2021)), so T3 inherits their efficiency and effectiveness thanks to this special instantiation capability.

**Natural regularization effect.** $\min \mathcal{L}(\mathbf{A}) \Rightarrow \min \mathcal{L}(\sum_{s=1}^{S} \|\mathbf{A} \odot \mathbf{M}_s\|_F^2)$ The window mechanism

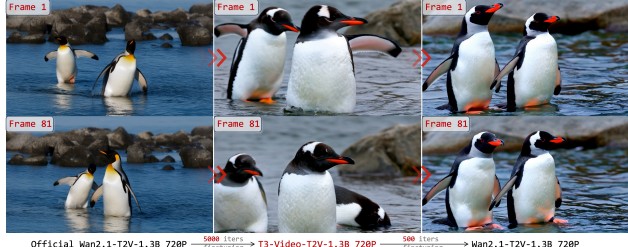

*Figure 4.* T3-Video restores full-attention capability through the re-transform attention process, which is potentially applicable to efficient pre-training of new architectures.

enforces attention weights to be zero outside windows, *i.e.*, $\mathbf{A}_s = \mathbf{A}_{\text{full}} \odot \mathbf{M}_s$, which is equivalent to an $\ell_0$ norm constraint. Since the $\ell_0$ norm is non-differentiable, we adopt the Frobenius norm as a convex relaxation. This constraint reduces the number of non-zero elements in the attention matrix from $N^2$ to $(n_t n_h n_w) \cdot m^2$, where $m = m_t \times m_h \times m_w$ is the number of tokens per window, and $(n_t n_h n_w)$ is the total number of windows. According to statistical learning theory, the generalization error bound is proportional to the square root of the effective number of parameters. The window mechanism reduces the effective parameters from $\mathcal{O}(N^2)$ to $\mathcal{O}(Nm)$, tightening the generalization bound: $\mathcal{R}_{\text{sparse}} \leq \mathcal{R}_{\text{full}} \cdot \sqrt{\frac{m}{N}}$. This structured sparsity naturally limits the model capacity, serving as a regularization mechanism that thereby enhances the model's generalization and reducing overfitting risks.

**Conceptually structure-aware adaption.** Compared with full fine-tuning that only changes parameters, T3 additionally performs structural fine-tuning. Its mechanism is similar to LoRA (Hu et al., 2022) and ControlNet (Zhang et al., 2023): the fundamental architecture remains unchanged and only undergoes slight modifications, inheriting weights and adapting the base model's capabilities through fine-tuned parameters. Therefore, its "only transform the Transformer" mechanism does not impair the model's learning ability or performance; rather, the local window attention inherently provides a form of regularization that suits the sparsified, redundant characteristics of video modeling.

**Feed back to naive full-attention for efficient training.** Recognizing T3 and full-attention as essentially the same model family, we explore using T3 to bootstrap full-attention computation. While training a 4K video-generation model from scratch is computationally expensive, T3-Video dramatically reduces resource needs and accelerates training. Pretrained T3-Video weights can then initialize a full-attention model for brief fine-tuning. We validate this by initializing at 720P with T3-Video, then fine-tuning with naive DiT; performance recovers after just 500 iterations (Fig. 4 and Tab. 7). Considering differences in training compute, strategies, and data, observed discrepancies are within error margins.

*Table 1.* **Disassembly of parameters and MACs between Wan2.1-T2V-1.3B (Top) and T3-Video-T2V-1.3B (Bottom)** that focuses on the last two columns ("Attn" and "ALL") of each row. Both of them contain same parameters. "Rest" includes text and time-related embeddings and cross-attention.

| | Resolution | Encoder | Decoder | DiT | | | | | |
| --- | --- | --- | --- | --- | --- | --- | --- | --- | --- |
| | | | | Rest | $QKV_{proj.}$ | $O_{proj.}$ | FFN | Attn | All |
| | Param. | 53.6M | 73.3M | 309.6M | 212.5M | 70.8M | 826.1M | 0 | 1419.0M |
| MACs | 480×832 | 81.2T | 137.0T | 4.7T | 7.0T | 2.3T | 27.1T | 98.9T / 4.5T×22.0↑ | 140.0T / 45.5T |
| | 720×1280 | 187.5T | 316.3T | 10.8T | 16.1T | 5.4T | 62.4T | 526.7T / 17.0T×30.9↑ | 621.4T / 111.7T |
| | 1088×1920 | 425.0T | 716.9T | 24.5T | 36.4T | 12.1T | 141.5T | 2706.2T / 77.5T×34.9↑ | 2920.7T / 291.9T |
| | 2176×3840 | 1699.9T | 2867.5T | 97.7T | 145.5T | 48.5T | 566.0T | 43299.3T / 1006.8T×43.0↑ | 44157.1T / 1864.5T |

*Table 2.* **Inference latency analysis** of T3-Video-1.3B (denoising 50 steps with CFG) and deployment version described in Sec. 2.5 (denoising 8 steps without CFG and along with *e*VAE). Unit: s. Top: Official. Bottom: T3-Video. ×↑ and ×↑ denote the relative speedups in the vertical and horizontal directions, respectively.

| Resolution | T2V-1.3B | | | T2V-1.3B-Deployment | | |
| --- | --- | --- | --- | --- | --- | --- |
| | Decoder | DiT (50) | Latency | eVAE | DiT (8) | Latency |
| 480×832 | 5.8 | 131.1 | 267.9 | - | - | - |
| | 5.8 | 50.4×2.6↑ | 106.6 | 0.233×24.9↑ | 4.0×12.5↑ | 4.3×25.0↑ |
| 720×1280 | 13.8 | 572.1 | 1,157.9 | - | - | - |
| | 13.8 | 123.0×4.7↑ | 259.9 | 0.514×26.9↑ | 9.8×12.5↑ | 10.4×25.1↑ |
| 1088×1920 | 49.8 | 2,653.7 | 5,357.2 | - | - | - |
| | 49.8 | 310.3×8.6↑ | 670.4 | 2.056×24.2↑ | 24.8×12.5↑ | 26.9×24.9↑ |
| 2176×3840 | 451.0 | 39,661.7 | 79,774.4 | - | - | - |
| | 451.0 | 1,857.4×21.4↑ | 4,165.8 | 18.176×24.8↑ | 148.6×12.5↑ | 166.8×25.0↑ |

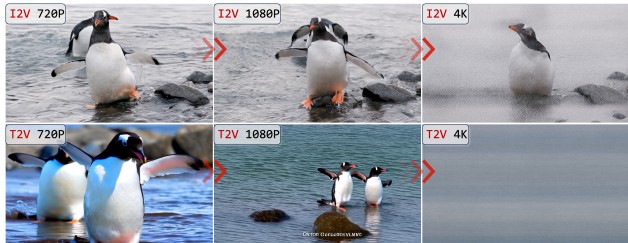

*Figure 5.* Directly scaling T3-Video-1.3B from 720P leads to performance degradation, which cannot adapt to arbitrary resolution inference. For I2V, the degradation is alleviated due to image prior.

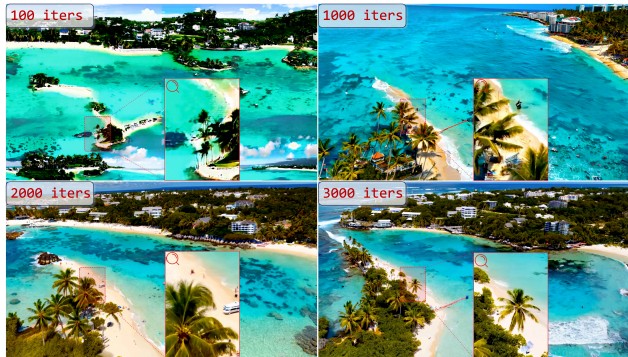

*Figure 6.* Directly fine-tuning T3-Video-T2V-1.3B for native 4K generation: as training progresses, the model first adapts to the spatial structure and then progressively refines the fine details.

## 2.4. Achieving Native 4K Video Generation

**High-efficient parameter-inherited finetuning.** We adopt Wan2.1-I2V 1.3B as the base model, transform it into T3-Video-I2V-1.3B, and inherit the official weights. This model is fully fine-tuned (all parameters) at 720P. Directly altering the model's inference resolution causes generation failures (Fig. 5); therefore, the resulting weights are then used to fully fine-tune a 4K (2176×3840) model. Compared to directly fine-tuning a 4K video-generation model from the official weights, this progressive scheme markedly accelerates convergence by only using 500 iterations.

**Compatible with resolution-aware LoRA adaption.** From the perspective of practical application costs, we further accelerate inference of T3-Video with step and CFG distillation; using 4K resolution directly poses severe challenges to GPU resources (primarily compute and memory). Therefore, we further applied LoRA to the T3-Video model that had been fine-tuned at 720P for high-resolution adaptation, and similarly observed high-quality 4K video generation, demonstrating the effectiveness of the approach (see Sec. 3.2.2). Tab. 5 presents the quantitative experimental results; compared with full-parameter fine-tuning, T3-Video-T2V-1.3B-LoRA exhibits only a slight decrease in performance but is still superior to the comparison methods.

**Efficiency analysis.** *i)* Tab. 1 further analyzes the MACs distribution of T3-Video versus the base Wan2.1-T2V-1.3B across different resolutions under the same parameter count. The re-

*Table 3.* Memory analysis.

| Reso. | Train | Test |
| --- | --- | --- |
| 720P | 29.6G | 8.8G |
| 1080P | 52.8G | 18.6G |
| 4K | 179.4G | 59.5G |

sults show that T3 achieves a substantial theoretical reduction in MACs compared to the baseline, up to 43.0× at 4K resolution. *ii)* We also performed an inference latency analysis as shown in Tab. 2: acceleration factors of 2.6/4.7/8.6/21.4× were observed at 480P/720P/1080P/4K, respectively. Although these observed speedups do not fully reach the theoretical values, the gap is due to hardware limitations and the current lack of joint optimization between hardware and framework. *iii)* In addition, we report training and inference memory usage in Tab. 3; inference on 4K with 81 frames requires less than 60G of memory.

**Analysis of the Learning Process.** Fig. 6 illustrates the evolution of generated outputs during the training of our T3-Video. By 2,000 iterations it already produces satisfactory results, demonstrating that T3-Video converges rapidly.

## 2.5. Efficient Inference Deployment

To further explore the applicability of $T_3$ in high-resolution video generation, we empirically demonstrate that it can be adapted to several non-structural acceleration schemes.

**Compatible with Step and CFG Distillation.** By default we use 50 steps for inference, which still imposes substantial computational demands. Therefore, based on DMD2 (Yin et al., 2024), we perform simultaneous 8-step and CFG distillation on a small amount of data to equip T3-Video, without using a GAN loss. This procedure can greatly ac-

*Table 4.* **Efficiency and performance of efficient *e*VAE over official Wan2.1-1.3-VAE and Wan2.2-5B-VAE for faster inference.** Defalut 720×1280 resolution on one H20 GPU.

| VAE | Encoder | | Decoder | | | | PSNR | SSIM | LPIPS |
|---|---|---|---|---|---|---|---|---|---|
| | Params. | MACs | Params. | MAC | Latency | Speedup | | | |
| Wan2.1-1.3B | 53.60M | 187.49T | 73.30M | 316.26T | 13.8380 | 1.0× | 38.07 | 0.9576 | 0.0251 |
| *e*VAE-Wan2.1-1.3B-10M | 1.47M | 5.86T | 9.84M | 13.18T | 0.5145 | 26.9× | 36.29 | 0.9422 | 0.04 |
| Wan2.2-5B | 149.64M | 130.82T | 555.05M | 688.58T | 10.5796 | 1.0× | 38.30 | 0.9567 | 0.0324 |
| *e*VAE-Wan2.2-5B-35M | 149.64M | 130.82T | 34.97M | 43.34T | 1.3040 | 8.1× | 37.14 | 0.9484 | 0.052 |

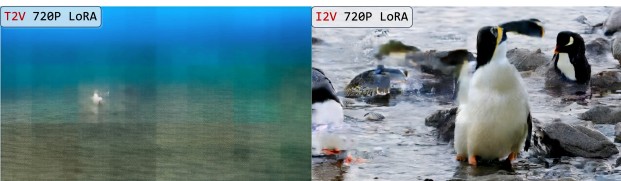

*Figure 7.* Formidable direct LoRA may fail.

celerate the model's inference by 12.5× with negligible accuracy degradation, as shown in Tab. 2.

**Compatible with Advanced Efficient VAE.** As the fraction of computation spent on attention decreases, the VAE decoder remains large in scale and its share of inference cost increases, which will become an application bottleneck. Therefore, we improve and train efficient versions termed *e*VAE. As shown in Tab. 4, *e*VAE-Wan2.1-1.3B-10M significantly reduces the decoder parameter count and MACs from 73.3M/316.26T to 9.84M/13.18T, while the reconstructed LPIPS increases only from 0.0251 to 0.04, which is below the human-perceptible threshold of 0.05.

### 2.6. Bottlenecks and Promising Optimizations

**Unsatisfactory direct LoRA-tuning from official weight.** Fig. 2 shows that window attention modeling remote token interactions produces noise during direct inference, revealing a large gap between transformed structure and original features that hinders learning. We analyze direct LoRA fine-tuning effects on T3-Video-T2V-1.3B (Wang et al., 2025a) and T3-Video-I2V-1.3B (Team, 2025a). As shown in Fig. 7, T3-Video-T2V-1.3B (Wang et al., 2025a) fails to learn, while T3-Video-I2V-1.3B (Team, 2025a), with its first-frame reference-domain constraint, partially preserves first-frame semantics but still produces artifacts and blockiness. LoRA ranks 32, 64, and 128 all yielded poor results. This motivated our approach in Sec. 2.6: first fully fine-tune at low resolution for T3-domain alignment, then apply LoRA for high-resolution adaptation.

**Challenge on training data and computational power.** This paper preliminarily explores lightweight video model architectures; however, limited by data (42K Ultra-Video (Xue et al., 2025)) and compute (batch size 64), we didn't investigate larger models (*e.g.*, Wan2.1-14B), which would likely improve semantic consistency and visual quality.

**Gap between actual and theoretical speed.** As shown in Tab. 1, DiT's theoretical 43.0× MACs speedup at 4K yields

only 21.4× actual inference acceleration (Tab. 2) due to software-hardware mismatch. This gap widens at lower resolutions: at 720P, theoretical 30.9× speedup achieves only 4.7× actual speedup, caused by reduced compute density from tiling. However, higher resolutions and better software-hardware co-optimization offer significant improvement potential. **Mixed-resolution training.** Existing models are trained only at a fixed resolution, but training with mixed resolutions can further enhance a model's applicability by allowing flexible reconfiguration according to application requirements; additionally, different scales can potentially complement one another to improve overall performance.

**More appropriate evaluation metrics.** Since existing perceptual models only accept inputs below 512 × 512 resolution and cannot adequately assess generated video quality, we excluded VBench (Huang et al., 2024) metrics, which also caused out-of-memory errors. High-resolution video generation urgently requires research on high-resolution perceptual foundation models and corresponding adaptations.

**Potential cascaded super-resolution pipeline.** Cascaded pipelines combining low-resolution video generation and subsequent super-resolution methods are generally inferior to end-to-end pipelines in terms of efficiency or quality under comparable parameter scales and computational costs. Their limitations lie in redundant feature extraction and error accumulation. For 4K video generation, high-frequency details produced during super-resolution upsampling often lack semantic coherence, resulting in temporal flickering and visual distortion. Experiments conducted on Ultra-Gen (Hu et al., 2026) verify that native generation substantially outperforms cascaded pipelines in video quality and text-image alignment, as it avoids the inherent semantic fragmentation issue of cascaded schemes. This paper focuses on native 4K video generation scenarios, and its core argument is that T3-Video surpasses existing native methods in both efficiency and quality for end-to-end native 4K video generation. We do not oppose the cascaded low-resolution plus super-resolution pipeline, which still retains advantages in scenarios with limited GPU memory or cross-team collaboration demands, such as one team generating videos for enhancement by another team.

**Compatibility.** T3-Video does not alter the attention computation in a window, and can naturally be compatible with various attention improvements (sparse attention that relies on spatial-temporal redundancy, *e.g.*, SVG (Xi et al., 2025)/STA (Zhang et al., 2025h)/RadialAttention (Li et al., 2026)) and dynamic routing methods like MoGA (Jia et al., 2026) that integrate multiple designs.

*Table 5.* Comparison with SoTAs on native 4K Video generation with pretrained models from UltraGen (Hu et al., 2026).

| Model | VQA | VTC | DoG | BM | RA | TDS | TEP |
|---|---|---|---|---|---|---|---|
| Wan2.1-T2V-1.3B (Official) | 30.01 | 0.37 | 0.22 | 0.968 | 0.76 | 0.83 | 0.33 |
| HunyuanVideo (Official) | 61.92 | 0.68 | 0.26 | 0.961 | 0.72 | **0.91** | 0.34 |
| UltraGen (Hu et al., 2026) | 67.43 | 0.75 | 0.45 | 0.980 | 0.72 | 0.84 | 0.52 |
| T3-Video-T2V-1.3B (Ours) | **71.72** | **0.83** | **0.54** | **0.988** | **0.83** | **0.91** | **0.70** |
| T3-Video-T2V-1.3B-LoRA (Ours) | 70.78 | 0.79 | 0.50 | 0.990 | 0.79 | 0.91 | 0.61 |

*Table 6.* Multi-baseline generalization of T3-Video (4K).

| | Model | VQA | VTC | DoG | BM | RA | TDS | TEP |
|---|---|---|---|---|---|---|---|---|
| I2V | Wan2.1-I2V-1.3B (Official) | 59.75 | 0.79 | 0.27 | 0.974 | 0.65 | 0.82 | 0.32 |
| | T3-Video-I2V-1.3B (Ours) | 63.6 | 0.82 | 0.32 | 0.986 | 0.83 | 0.92 | 0.41 |
| T2V | Wan2.2-T2V-5B (Official) | 47.23 | 0.52 | 0.16 | 0.984 | 0.51 | 0.85 | 0.25 |
| | T3-Video-T2V-5B (Ours) | 67.40 | 0.91 | 0.35 | 0.995 | 0.82 | 0.88 | 0.34 |
| I2V | Wan2.2-I2V-5B (Official) | 65.13 | 0.89 | 0.28 | 0.989 | 0.67 | 0.86 | 0.38 |
| | T3-Video-I2V-5B (Ours) | 68.84 | 0.90 | 0.34 | 0.996 | 0.84 | 0.90 | 0.44 |

*Table 7.* Empirical observations on basic factors (720P).

| | Model | VQA | VTC | DoG | BM | RA | TDS | TEP |
|---|---|---|---|---|---|---|---|---|
| (a) Return Official | Wan2.1-T2V-1.3B (Official) | 70.56 | 0.91 | 0.42 | 0.938 | 0.71 | 0.87 | 0.51 |
| | T3-Video-T2V-1.3B (Ours) | 69.37 | 0.90 | 0.40 | 0.948 | 0.72 | 0.89 | 0.49 |
| | Wan2.1-T2V-1.3B (Return Ours) | 69.51 | 0.90 | 0.42 | 0.925 | 0.71 | 0.88 | 0.49 |
| (b) Batch Size | 8 | 66.65 | 0.81 | 0.30 | 0.928 | 0.67 | 0.88 | 0.47 |
| | 16 | 67.17 | 0.83 | 0.39 | 0.924 | 0.66 | 0.88 | 0.42 |
| | 32 | 68.05 | 0.84 | 0.36 | 0.920 | 0.71 | 0.88 | 0.49 |
| | 64 | 69.37 | 0.90 | 0.40 | 0.948 | 0.72 | 0.89 | 0.49 |
| | 128 | 68.72 | 0.90 | 0.41 | 0.923 | 0.74 | 0.90 | 0.51 |
| | 256 | 68.43 | 0.89 | 0.41 | 0.926 | 0.74 | 0.87 | 0.49 |
| (c) Local Type | Wan2.1-T2V-1.3B (Swin) | 67.34 | 0.87 | 0.33 | 0.916 | 0.65 | 0.86 | 0.47 |
| | T3-Video-T2V-1.3B (Ours) | 69.37 | 0.90 | 0.40 | 0.948 | 0.72 | 0.89 | 0.49 |
| (d) Layer Config. | T3-Video-T2V-1.3B (Large Ratio) | 67.14 | 0.88 | 0.38 | 0.900 | 0.69 | 0.87 | 0.47 |
| | T3-Video-T2V-1.3B (Small Ratio) | 68.69 | 0.88 | 0.42 | 0.913 | 0.68 | 0.90 | 0.46 |
| | T3-Video-T2V-1.3B (Ours) | 69.37 | 0.90 | 0.40 | 0.948 | 0.72 | 0.89 | 0.49 |
| (e) Light-Weight | T3-Video-T2V-1.3B (Deployment) | 67.72 | 0.85 | 0.36 | 0.891 | 0.63 | 0.86 | 0.45 |
| | T3-Video-T2V-1.3B (Ours) | 69.37 | 0.90 | 0.40 | 0.948 | 0.72 | 0.89 | 0.49 |

*Table 8.* VBench evaluation results per dimension. $^*$: Videos are downsampled to 1K to avoid OOM.

| Models | Subject Consistency | Background Consistency | Temporal Flickering | Motion Smoothness | Dynamic Degree | Aesthetic Quality | Imaging Quality | Object Class |
|---|---|---|---|---|---|---|---|---|
| UltraWAN-4K (LoRA, 29f) | 96.05% | 98.02% | 98.88% | 98.47%* | 66.66%* | 56.81% | 71.61% | 50.00% |
| T3-Video-4K (FT, 81f) | 97.17% | 98.10% | 98.52% | 98.85%* | 66.66%* | 59.83% | 72.18% | 66.66% |

| Models | Multiple Objects | Human Action | Color | Spatial Relationship | Scene | Appearance Style | Temporal Style | Overall Consistency |
|---|---|---|---|---|---|---|---|---|
| UltraWAN-4K (LoRA, 29f) | 42.75% | 66.66% | 100.0% | 100.0% | 00.00% | 19.46% | 19.31% | 22.88% |
| T3-Video-4K (FT, 81f) | 45.62% | 66.66% | 100.0% | 100.0% | 16.66% | 19.28% | 21.15% | 23.61% |

# 3. Experiments

## 3.1. Implementation Details

**Baselines.** We use text-to-image versions of Wan2.1-1.3B, Wan2.2-5B, and HunyuanVideo-13B as base models, with UltraWan (Xue et al., 2025) and UltraGen (Hu et al., 2026) as direct baselines (both trained on 29-frame 4K videos). Since UltraGen (Hu et al., 2026) already proved superior to cascaded low-resolution plus super-resolution approaches, we omit that comparison. We select Wan2.1-T2V-1.3B as the default basic model.

**Datasets.** We use the open-sourced UltraVideo (Xue et al., 2025) dataset (42,184 videos) for fine-tuning, as it provides high-quality videos with detailed captions and high-resolution data. We randomly select 120 short videos as 4K-VBench and use the remaining 42K videos for training, adopting UltraWan's (Xue et al., 2025) random caption sampling strategy.

**High-resolution video assessment.** We evaluate methods using 4K-VBench through *video quality assessment* (VQA) at 1080P and 4K with FineVQ (Duan et al., 2025), and *video-text consistency* (VTC) using Qwen3-VL-32B (Team, 2025c; Bai et al., 2025). We introduce metrics for high-resolution details: *1)* DoG, *2)* BM, and *3)* RA for *spatial details*; *4)* TDS and *5)* TEP for *temporal details*. A human study evaluates quality aesthetics, textual consistency, and detail quality.

**Training details.** T3, a generic video model architecture improvement, is evaluated using Wan family-based models. Models are trained on UltraVideo using AdamW optimizer for 5K iterations (batch size 64) on H20 GPUs with Deep-Speed ZeRO stage 2. Learning rates are 2e-5 (full finetuning) and 1e-4 (LoRA, rank=64). Base models are pretrained and ablated at 720P (720×1280), with high-resolution fine-tuning at 1080P (1088×1920) and 4K (2176×3840). All experiments use DiffSynth-Studio (Team, 2024).

## 3.2. Experimental Results

### 3.2.1. T3-VIDEO *vs.* NATIVE 4K VIDEO METHODS

Tab. 5 shows that T3-Video achieves clear quantitative superiority over the SoTAs, with particularly notable improvements relative to the recent UltraGen (Hu et al., 2026). We also provide a qualitative analysis in Fig. 1 demonstrating that our T3-Video better handles semantically complex scenes (including moving subjects such as animals and humans, which are harder to generate) while also exhibiting a clear efficiency advantage over competing methods.

### 3.2.2. RESULTS ON VARIOUS BASE MODELS

With limited resources, we further scale T3-Video to the larger Wan2.2-5B and, for the first time, extend it to 4K I2V generation as shown in Tab. 6; we likewise observe a consistent improvement in efficiency and superior performance over baselines. Fig. A1 in Sec. B compares results across models of different scales under different training settings, and similarly finds that our T3-Video can be effectively extended to LoRA fine-tuning, as discussed in Sec. 2.6.

### 3.2.3. EMPIRICAL OBSERVATIONS AND ANALYSIS

We further analyze T3-Video in Tab. 7.
**Feed back to naive full-attention.** As discussed in Sec. 2.6, T3 and full-attention are essentially equivalent in that T3-Video can be transformed back into full-attention, and (a) demonstrates the effectiveness of this equivalence.
**Batch size.** Larger sizes do not bring noticeable positive gains that may be due to the amount of data, whereas smaller batch sizes result in significantly worse performance (b).

*Table 9.* Human study with UltraGen (Hu et al., 2026). ① Video Quality, ② Text Consistency, ③ Temporal Consistency, and ④ Detail Richness.

| Method | ① | ② | ③ | ④ |
|---|---|---|---|---|
| UltraGen (Hu et al., 2026) | 28.75% | 40.25% | 43.08% | 36.58% |
| Ours | 71.25% | 59.75% | 56.92% | 63.42% |

**Local type.** We also replaced full-attention with Shifted Windows (Liu et al., 2021) as discussed in Sec. 2.2, and T3-Video still exhibits a clear advantage (c).

**Layer configure.** The hierarchical ratio is critical for T3-Video, as discussed in Sec. 2.2. We additionally designed two configurations with block ratios as large as possible ($3 <$ ratio $< 6$) and as small as possible ($1 <$ ratio $< 3$); the results in (d) indicate that covering a more diverse set of T3-Video instances via the ratio yields better performance (see '*diffsynth/models/wan_video_dit.py* (Line-160)' in the attached source code).

**Light-weight deployment.** Results in (e) demonstrate the effectiveness of the lightweight acceleration scheme described in Sec. 2.5, showing that the quantitative drop in performance metrics is within an acceptable range relative to the achieved speedup.

**Results on VBench with UltraWAN.** Tab. 8 presents the comparison results between T3-Video and UltraWAN at 4K resolution on the Vbench (Huang et al., 2024), achieving superior performance with an order-of-magnitude speedup.

**Human study with UltraGen.** We recruited 10 professional video evaluators to conduct a win-rate evaluation between UltraGen and T3-Video-T2V-1.3B for 4K video generation. Results in Tab. 9 demonstrate that our method exhibits a significant human preference compared to the baseline method.

## 4. Conclusion

This paper tackles efficient 4K video generation through T3-Video, an architecture-adaptation framework for pretrained Transformers. Using multi-scale shared-window attention and hierarchical optimization, T3-Video achieves linear computational scaling with resolution while preserving pre-trained weights to reduce re-training costs. Results show T3-Video surpasses state-of-the-art methods in quality and efficiency, generating 81-frame 4K videos with $10\times$ speedup versus official models, offering a practical high-resolution synthesis solution.

**Limitation and future work.** Training was limited by the 42K UltraVideo dataset and computational constraints, preventing larger-scale model exploration; investigating mixed-resolution training could improve adaptability; future work will focus on hardware-software co-optimization and extending T3-Video to minute-scale 4K generation with enhanced temporal consistency.

## Acknowledgments and Disclosure of Funding

This work is supported by the National Key R&D Program of China (Grant No. 2025YFF0511302) and National Natural Science Foundation of China (No. 625B2115).

## Impact Statement

This research introduces **T3-Video**, a framework that optimizes the attention logic of pretrained Transformers to significantly lower the computational barriers for native 4K video generation. While driving progress in fields such as media arts, film production, and virtual reality, this technology carries notable social implications and ethical considerations:

1. **Democratization of Technology and Resource Efficiency**: T3-Video accelerates native 4K video generation by more than 10x and enables the generation of 81-frame videos with less than 60G of memory on a single GPU. By reducing reliance on massive computing clusters, it allows individual creators and smaller institutions to develop ultra-high-definition content at a much lower cost.

2. **Content Authenticity and Deepfake Risks**: As the quality of native 4K video improves significantly, generated content becomes increasingly indistinguishable from real footage in terms of fine texture and semantic consistency. This may heighten the difficulty of identifying "Deepfake" content, potentially misleading public perception of real-world events. We advocate for the use of digital watermarking or metadata provenance alongside this technology to ensure content transparency.

3. **Copyright and Data Ethics**: The framework relies on fine-tuning with large-scale video datasets like Ultra-Video. This underscores the importance of establishing transparent, legal data acquisition mechanisms and respecting original copyrights in the era of artificial intelligence.

4. **Computational Efficiency and Environmental Impact**: By reducing computational complexity from quadratic to approximately linear , T3-Video decreases the energy consumption per unit of generation. Such algorithmic optimizations help mitigate the carbon footprint and environmental pressure associated with training large-scale AI models.

In summary, T3-Video is a neutral productivity tool designed to enhance ultra-high-definition content creation. We encourage developers and users to explore its potential within legal and ethical frameworks to build a responsible AI ecosystem.

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

# Appendix

**Overview**

The appendix presents the following sections to strengthen the main manuscript:

— We provide a demo video and source code in the Supplementary Materials for a better understanding of our work.

— **Sec.** A shows the Related Work part of the paper.

— **Sec.** B shows qualitative results across different scales and training settings.

— **Sec.** C shows metric details in 4K-VBench.

# A. Related Work

## A.1. Video Generation Foundation Models

The development of diffusion techniques has substantially improved image generation quality, as exemplified by the SD (Rombach et al., 2022; Podell et al., 2024; Esser et al., 2024) and FLUX (Labs, 2024a; Labs et al., 2025) families. These advances were subsequently extended to video generation, giving rise to works such as AnimateDiff (Guo et al., 2024) and SVD (Blattmann et al., 2023a), which drive progress by expanding temporal modules. Transformer-based DiT architectures have become a standard component in subsequent video-generation models; in particular, Sora demonstrated a breakthrough in photorealistic video synthesis, after which both open-source (Chen et al., 2025b; HaCohen et al., 2024; Ma et al., 2025; Chen et al., 2025a; Kong et al., 2024; Wang et al., 2025a) and closed-source (Bao et al., 2024; Labs, 2024b; Research, 2023; Limited, 2025; Gao et al., 2025; Zhang et al., 2025j; Technology, 2024; Teng et al., 2025; Wiedemer et al., 2025; Team, 2025b) models have seen continuous improvements. Wan2.1/2.2 (Wang et al., 2025a) and HunyuanVideo (Kong et al., 2024) have attracted particular attention for their strong results and well-maintained open ecosystems, and they have been widely used in downstream tasks. Other efforts have focused on high-resolution and long-term video generation by introducing autoregressive modeling or improved attention mechanisms. Considering both performance and efficiency, this work adopts Wan2.2-5B (Wang et al., 2025a) as the base video model.

## A.2. High-Resolution Video Generation

High-resolution image generation (Zhang et al., 2025c; Yu et al., 2025) has emerged as an important research direction for media applications. Existing video-generation work, however, typically targets 480P or 720P; few methods support native 1K/2K video generation, and native 4K generation remains largely infeasible. LinGen (Wang et al., 2025b) leverages linear Mamba2 (Dao & Gu, 2024) to generate minute-long videos, but increasing spatial resolution leads to a token count that grows by orders of magnitude relative to the temporal axis, and training at 512P already consumed on the order of 10K Nvidia H100 GPU days that is an enormous resource requirement. To mitigate this, several works (Blattmann et al., 2023b; Wang et al., 2025c; Zhang et al., 2025i) adopt a low-resolution generation followed by video super-resolution (Zhou et al., 2024), but this pipeline is complex and the added high-frequency details often lack semantic content, improving only perceived sharpness while still failing to achieve high-quality 4K synthesis. Very recently, UltraVideo (Xue et al., 2025) introduced the concept of 4K video generation and released the first open 4K video dataset; their finetuned UHD-4K UltraWan (Xue et al., 2025) explored native 4K generation but did not address efficiency. UltraGen (Hu et al., 2026) improves model efficiency for 4K but relies on additional architectural finetuning and yields suboptimal quality. In this work, we introduce a multirole window attention mechanism to optimize the inference logic of a pretrained Transformer, enabling efficient fine-tuning and optimization for 4K video models using only modest computational resources.

## A.3. Efficient Video Generation

Efficient video inference and deployment are critical. TeaCache (Liu et al., 2025b) accelerates diffusion models via timestep-aware input caching, while KV-Cache achieves speedups through key-value caching. In the video domain, the FlashAttention (Dao et al., 2022; Dao, 2023; Shah et al., 2024) and SageAttention (Zhang et al., 2025e;b;d) families are widely used as training-free, drop-in attention replacements; SpargeAttention (Zhang et al., 2025f) further proposes train-free sparse attention to speed up inference. SVG (Xi et al., 2025) relies on online analysis to differentiate attention heads; STA (Zhang et al., 2025h) focuses on sliding block kernel optimization; RadialAttention (Li et al., 2026) uses energy decay

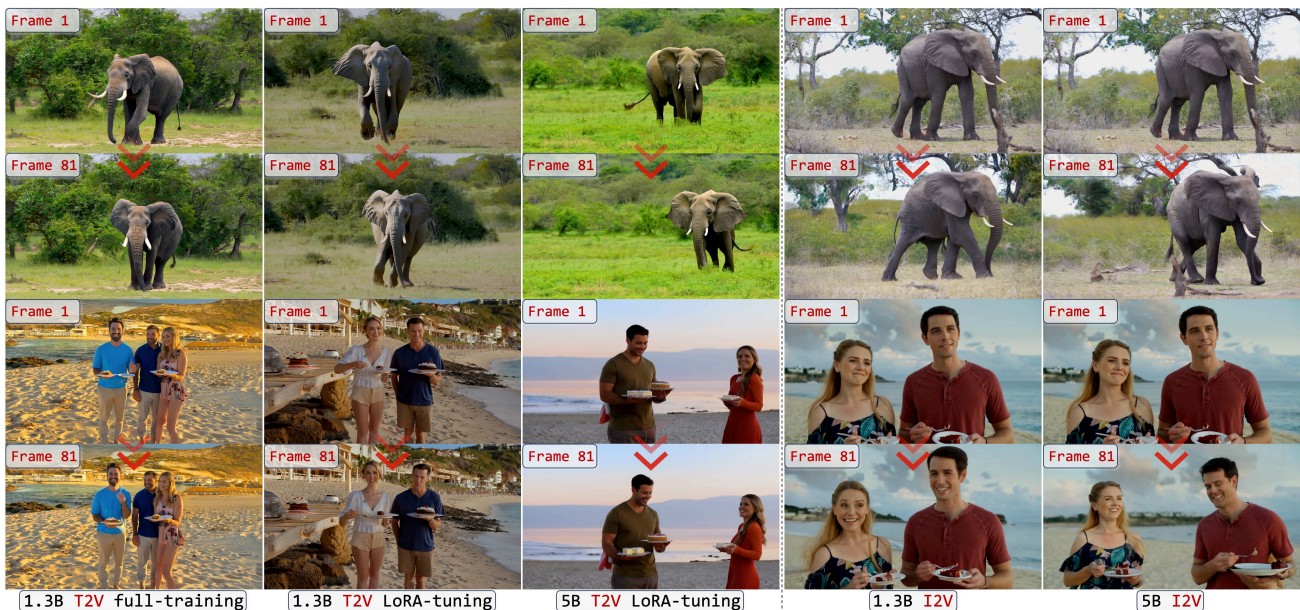

*Figure A1.* T3-Video series (T2V and I2V) based on Wan2.1-1.3B and Wan2.2-5B achieve satisfactory results in both full-/LoRA-tuning.

masks. These methods target training-free or lightweight fine-tuning for acceleration. On the other hand, VSA (Zhang et al., 2025g) introduces trainable sparse attention, and FPSAttention (Liu et al., 2025a) incorporates FP8 quantization for joint training, both yielding substantial inference gains. In this work we adopt the attention-transparent FlashAttention2 (Dao, 2023) as a safe, non-invasive replacement and deliberately do not pursue quantization or sparse-attention variants that would significantly alter the original attention structure. For deployment efficiency, we also introduce step and CFG distillation techniques inspired by (Yin et al., 2024), and design lightweight versions of the pretrained encoder and decoder to better match the acceleration achieved in the DiT backbone.

## B. Qualitative Results across Different Scales and Training Settings

Fig. A1 in Sec. B compares results across models of different scales under different training settings, and similarly finds that our T3-Video can be effectively extended to LoRA fine-tuning, as discussed in Sec. 2.6.

## C. Metric Details in 4K-VBench

**FinvVQ for video quality assessment.** We use the Overall Score from FineVQ (Duan et al., 2025), which is based on InternVL2-8B (Chen et al., 2024), to evaluate the quality of generated videos.

**Qwen3-VL-32B for video-text consistency assessment.** We use open-sourced Qwen3-VL-32B (Team, 2025c; Bai et al., 2025) (closed-source solutions may experience changes in functionality as versions and engineering interfaces are updated.) as the MLLM to assess video–text consistency with the following prompt:

```
You are an expert evaluator for **text-to-video alignment**.

Your task is to objectively assess how well a generated video matches the given text description.

**Text Description:** {text_description}

### Evaluation Criteria
Carefully evaluate the degree to which the video aligns with the text description based on the following key
        aspects:
- **Scene & Environment** (location, background, objects, lighting)
- **Entities & Appearance** (people, animals, objects, and their visual characteristics)
- **Actions & Dynamics** (movements, activities, or physical events)
- **Temporal Consistency** (sequence, timing, and continuity of events)
- **Atmosphere or Emotion** (mood, tone, or implied emotion consistent with the text)

### Rating Scale
Give a single score from **1 to 10**, where:
- **10** = Perfectly matches the text in all major aspects.
```

```
- **9** = Almost perfect, only very minor differences.
- **8** = Very good, most aspects align well with small inaccuracies.
- **7** = Generally consistent but missing some details.
- **6** = Fair, aligns in part but several inconsistencies exist.
- **5** = Partial alignment, multiple elements missing or inaccurate.
- **4** = Weak alignment, only a few correct aspects.
- **3** = Poor alignment, mostly inconsistent with the text.
- **2** = Barely related to the text.
- **1** = Completely unrelated to the text description.

### Output Format
Provide **only the final numeric score (1-10)**.
Do **not** include any reasoning, explanation, or additional text.
```

The final score is divided by 10 to normalize it to the range of 0-1.

**Difference of Gaussians (DoG).** When assessing high-resolution video quality, high-frequency information (such as texture, edges, and fine detail) most directly reflects perceived sharpness and realism. Therefore, extracting and analyzing high-frequency components is critical for measuring detail fidelity. However, details in high-resolution video span a wide range of scales — from very fine textures to larger structural elements. A single-scale high-pass filter typically captures only a limited frequency band and thus misses details at other scales, failing to fully characterize overall sharpness. To address this, we employ multi-scale Difference-of-Gaussian (DoG) filters to obtain band-pass responses across different frequency ranges. Each scale corresponds to a different level of detail: small scales emphasize fine texture and micro-contrast, while larger scales highlight structure and contour levels. By fusing the multi-scale responses, we obtain a scale-robust and more stable high-frequency representation that more comprehensively reflects the video's fidelity across multiple detail layers and local structures.

$$HFE_i = \frac{\mathbb{E}\left[R_i^2\right]}{\mathbb{E}\left[G_i^2\right] + \varepsilon},$$
$$R_i = \sum_{\sigma \in \{0.8, 1.6, 3.2\}} \left(\text{Gauss}(G_i, \sigma) - \text{Gauss}(G_i, 1.6\sigma)\right) \tag{A1}$$

In this equation, $G_i$ represents the $i$-th grayscale frame, and $R_i$ is the Difference-of-Gaussian (DoG) response, which is computed by subtracting two Gaussian-blurred versions of the image at different scales $\sigma$. Specifically, $R_i$ is obtained by subtracting the Gaussian blur at scale $\sigma$ from that at scale $1.6\sigma$, with $\sigma \in \{0.8, 1.6, 3.2\}$. The expectation $\mathbb{E}[\cdot]$ denotes the average energy, and $\varepsilon$ is a small constant added to prevent division by zero. This ratio quantifies the proportion of high-frequency content, with higher values indicating sharper images with more detailed textures.

**Blockiness Measurement (BM).** Block artifacts in generative video models typically arise from local generation discontinuities, *e.g.*, patch-based synthesis, windowed attention, or local convolutional decoders that fuse inconsistently at boundaries. These artifacts appear as regular square boundaries or grid-like patterns that introduce a subtle tearing effect. To quantify this, we compute the ratio of across-block differences to within-block differences for adjacent blocks (*e.g.*, 8×8 regions). When the across-block difference is significantly larger than the within-block difference, it indicates insufficiently smooth transitions at spatial boundaries and thus structural discontinuity. This metric sensitively captures block boundaries introduced by local generation mechanisms, allowing evaluation of local consistency issues in high-resolution generated video.

$$BLK_i = \frac{\bar{\Delta}_{\text{bd}}(G_i)}{\bar{\Delta}_{\text{in}}(G_i) + \varepsilon},$$
$$\bar{\Delta}_{\text{bd}}(G_i) = \text{mean} \left|\nabla G_i\right|_{\text{block edges}},$$
$$\bar{\Delta}_{\text{in}}(G_i) = \text{mean} \left|\nabla G_i\right|_{\text{inside blocks}} \tag{A2}$$

In this equation, $\nabla G_i$ represents the gradient of the image. The terms $\bar{\Delta}_{\text{bd}}(G_i)$ and $\bar{\Delta}_{\text{in}}(G_i)$ correspond to the average gradient magnitudes at the block edges and inside the blocks, respectively. The ratio $BLK_i$ quantifies the extent of blockiness in the image: higher values indicate more prominent block boundaries, making the block structure more noticeable.

**Ringing Artifact (RA).** Sharpening or super-resolution models often produce unreal "over-sharp" edges, *i.e.*, ringing artifacts. These pseudo high-frequency signals can give a false impression of clarity. To quantify such distortions, we

analyze the band-pass filtered responses in strong edge regions and compute the 95%-percentile overshoot ratio. When signal excursions exceed the range of natural luminance variation, we consider ringing to be present.

$$
\begin{aligned}
RING_i &= \frac{Q_{95}(|R_i|)}{Q_{95}(|L_i|) + \varepsilon}, \\
R_i &= \sum_{\sigma \in \{0.8, 1.6, 3.2\}} \left( \text{Gauss}(G_i, \sigma) - \text{Gauss}(G_i, 1.6\sigma) \right), \\
L_i &= \nabla^2 G_i
\end{aligned}
\tag{A3}
$$

In this equation, $R_i$ represents the Difference-of-Gaussian (DoG) response across multiple scales, and $L_i$ denotes the Laplacian of the image $G_i$. The operator $Q_{95}(\cdot)$ extracts the 95th percentile of the respective magnitude values, while $\varepsilon$ is a small constant added for numerical stability. The ratio $RING_i$ quantifies the strength of high-frequency oscillations around edges, where larger values correspond to more pronounced ringing or overshoot artifacts.

**Temporal Detail Stability (TDS).** Beyond spatial quality, temporal consistency is critical for perceived video stability. If a generative model is temporally unstable, high-frequency details will flicker or drift. The TDS metric aligns adjacent frames using Farneback optical flow and computes the temporal standard deviation on mid-to-high frequency components to measure temporal consistency. Larger standard deviation indicates greater temporal fluctuation of details.

$$
TDS_{\text{avg}} = 1 - \frac{\mathbb{E}_{x,y}\left[ \text{Std}_t(R_t(x,y)) \right]}{\mathbb{E}_{x,y}\left[ \text{RMS}_t(R_t(x,y)) \right] + \varepsilon}
\tag{A4}
$$

where $R_t(x,y)$ denotes the high-frequency response at pixel $(x,y)$ of frame $G_t$, and $\text{Std}_t(\cdot)$ and $\text{RMS}_t(\cdot)$ are the temporal standard deviation and root mean square, respectively, computed across the sequence of frames. The term $\varepsilon$ is a small constant added for numerical stability. This metric measures the temporal stability of high-frequency details: higher values indicate more consistent and less fluctuating fine structures across frames.

**Temporal Edge Persistence (TEP).** The temporal continuity of edge structures is a key indicator of structural stability in video. Poor temporal consistency causes edges to break or jitter. The TEP metric extracts Canny edges from adjacent frames and computes their intersection-over-union (IoU). Higher IoU indicates more stable, coherent edges over time; lower IoU signals edge drift or discontinuity. This measure effectively quantifies the temporal coherence of fine structural details in high-resolution video.

$$
TEP_{\text{avg}} = \frac{1}{N-1} \sum_{i=1}^{N-1} \frac{|\hat{E}_i \cap E_{i+1}|}{|\hat{E}_i \cup E_{i+1}| + \varepsilon},
\tag{A5}
$$

$$
TEP_{\text{avg}} = \frac{1}{N-1} \sum_{i=1}^{N-1} \frac{|\hat{E}_i \cap E_{i+1}|}{|\hat{E}_i \cup E_{i+1}| + \varepsilon}
\tag{A6}
$$

where $E_i$ and $E_{i+1}$ denote the edge maps of the $i$-th and $(i+1)$-th frames, respectively. The warped edge map $\hat{E}_i = \text{Warp}(E_i; \text{Flow}(G_i \rightarrow G_{i+1}))$ aligns the $i$-th frame to the $(i+1)$-th frame using optical flow. The term $\varepsilon$ is a small constant added for numerical stability. This metric evaluates temporal edge persistence, quantifying how well edge structures remain consistent across consecutive frames. Higher values indicate stronger structural continuity and more temporally stable motion boundaries.

