# OpenReview forum: "Transform Trained Transformer for Accelerating Native 4K Video Generation"
_ICML.cc/2026/Conference — ICML 2026 regular_

### Official Review · Reviewer_Dr7s · 2026-03-09

**Soundness:** 3
**Presentation:** 3
**Significance:** 3
**Originality:** 3
**Overall Recommendation:** 5
**Confidence:** 4

**Summary:**

This paper proposes T3-Video (Transform Trained Transformer), a method for accelerating native 4K video generation in Transformer-based models. Without modifying the core architecture or pretrained weights, the method converts global self-attention into multi-scale shared window attention, which preserves global semantic modeling while significantly reducing computational complexity. Experiments show that T3-Video enables native 4K video generation with 81 frames on a single GPU, achieving over 10× inference speedup while maintaining or improving generation quality.

**Compliance With Llm Reviewing Policy:**

Affirmed.

**Final Justification:**

This paper itself demonstrates a high level of completeness, and the authors have adequately addressed my questions.

**Key Questions For Authors:**

1. Training Stability and Hyperparameter Sensitivity. The paper introduces multiple technical contributions but lacks systematic ablation experiments to verify their individual effectiveness, as well as sensitivity under certain special parameters, such as window size and quantitative demonstration of layer configurations.
2. Compared with MoGA[1], which adopts a dynamic routing approach, how does it compare with the structured windows in this paper? Have you attempted similar strategies, and if not, why not?

[1] MoGA: Mixture-of-Groups Attention for End-to-End Long Video Generation

**Limitations:**

yes

**Strengths And Weaknesses:**

Soundness: The technical solution is reasonable, with complexity analysis provided and sufficient experimental evidence demonstrating effectiveness.

Presentation: The paper structure is clear, with charts and pseudocode aiding comprehension.

Significance: Effectively addresses the computational bottleneck in native 4K video generation, achieving 81 frames of 4K inference on a single GPU with significant speedup and quality improvement, demonstrating high engineering and application value.

Originality: While not proposing a new theoretical paradigm, it cleverly combines strategies such as s a multi-scale weight-sharing window attention, hierarchical blocking with an axis-preserving full-attention design, reflecting innovation in both methodology and engineering.

---

> ### Author Rebuttal · Authors · 2026-03-31
>
> **Q1: Training Stability and Hyperparameter Sensitivity**
>
> Thank you for your valuable question.
> 1) We conduct ablation studies on core designs including local attention type, layer configuration, and batch size in Table 7, verifying the independent effectiveness of our module. 2) We supplement experiments with larger/smaller window size (WS) and scale S=4 to extend Table 7 (the nearest resolution satisfying 720P is adopted since the default resolution is not divisible by 2). Results show that a larger WS slightly improves performance but increases attention MACs. 3) Our default layer configurations are optimal under GPU memory constraints, balancing generation quality and hardware practicality (L199), while Large/Small Ratio configurations degrade performance. 4) We will add sensitivity analysis on window size and hierarchical block ratio in the revised version to provide more references for real-world deployment.
> |Model|VQA|VTC|DoG|BM|RA|TDS|TEP|Attn MACs|
> |:-:|:-:|:-:|:-:|:-:|:-:|:-:|:-:|:-:|
> |WS=x2/S=2|69.93|0.91|0.41|0.942|0.72|0.88|0.50|56.7T|
> |WS=x0.5/S=2|68.12|0.85|0.36|0.925|0.69|0.87|0.46|4.7T|
> |WS=x1/S=4|69.45|0.90|0.41|0.953|0.72|0.89|0.47|36.5T|
> |T3-Video-T2V-1.3B (Default WS=x1/S=2)|69.37|0.90|0.40|0.948|0.72|0.89|0.49|17.0T|
>
> ---
> **Q2: Compared with MoGA**
>
> Thank you for your valuable comment.
> 1) MoGA uses learnable dynamic routing for token grouping, bringing extra forward computation overhead and training convergence costs. T3-Video adopts structured multi-scale shared windows, enabling zero-cost code-level replacement by reusing pre-trained weights without extra routing modules. It only modifies the forward logic of attention with one-line code replacement, which is simple and elegant, differing from the motivation of MoGA.
> 2) We refrain from dynamic routing to maximize parameter compatibility. In the revised version, we will analyze the applicable boundaries of static structures and dynamic routing in complex scenarios in related work to better distinguish different methods.
> 3) T3-Video does not alter the attention computation in a window, and can naturally be compatible with various attention improvements and MoGA-style methods that integrate multiple designs. We will add relevant discussions in Sec.2.6. Bottlenecks and Promising Optimizations of the revised version.

---

> > ### Author Rebuttal · Reviewer_Dr7s · 2026-04-03
> >
> > I have no more questions, I will improve my confidence.

---

> > > ### Author Response · Authors · 2026-04-03
> > >
> > > Dear Reviewer Dr7s:
> > >
> > > Thank you for recognizing our work. We commit to incorporating the content you suggested in the revised version. Thank you again for your effort in the review and the discussion!
> > >
> > > Best regards!
> > >
> > > Authors of T3-Video

---

### Official Review · Reviewer_Q5G9 · 2026-03-12

**Soundness:** 3
**Presentation:** 3
**Significance:** 3
**Originality:** 2
**Overall Recommendation:** 4
**Confidence:** 4

**Summary:**

This paper addresses efficient native 4K video generation for diffusion Transformers. The main idea is to retrofit pretrained full-attention video models with a transformed attention process, T3, which replaces global single-scale attention with multi-scale shared-window attention while largely preserving the original Transformer architecture and pretrained weights. The method is further combined with hierarchical blocking, axis-preserving attention patterns, progressive 720P-to-4K adaptation, LoRA-based high-resolution tuning, and a lightweight deployment stack based on distillation and eVAE. Empirically, the paper reports improved quality and efficiency on a proposed 4K benchmark, outperforming UltraGen and official baselines, and extending to both T2V and I2V settings.

**Compliance With Llm Reviewing Policy:**

Affirmed.

**Key Questions For Authors:**

- The discussion in Sec. 2.3 about regularization and generalization is interesting, but currently feels more intuitive than validated. Do the authors have empirical evidence that the sparse multi-scale window structure improves generalization beyond efficiency?

- The paper states that cascaded low-resolution plus super-resolution methods are omitted because UltraGen already showed superiority over that route. Could the authors summarize that comparison more explicitly in this paper, so that readers can better understand the intended scope of the empirical claim?

- Relatedly, should the main empirical claim of this paper be interpreted as “better than existing native-4K approaches,” or also as evidence against cascaded low-resolution-plus-super-resolution pipelines more broadly? I think the manuscript should state this more explicitly.

**Limitations:**

yes

**Strengths And Weaknesses:**

### Strengths
The paper tackles an important and practically relevant problem. Native 4K video generation is computationally challenging, and the proposed method directly targets the main bottleneck of full attention in long spatiotemporal sequences while remaining compatible with pretrained video Transformers. This is a practically meaningful direction, especially given the cost of training large video models from scratch.

From a technical standpoint, the method is plausible and the paper provides a reasonable amount of empirical support. The complexity argument is clear, and the ablations on local attention type, layer configuration, batch size, and return-to-full-attention behavior help justify the design choices. The main empirical results are strong enough to make the paper noteworthy, and the method appears to transfer beyond a single model setting.

The strengths of the paper do not mainly lie in novelty. The individual ingredients are not all new, and the theory section is more suggestive than rigorous. However, the paper’s value is in putting together a coherent and practically useful retrofit recipe for pretrained video Transformers, and in showing that this recipe yields meaningful gains for native 4K generation.

### Weaknesses
My main reservations are threefold. First, the theory section is not especially strong: the discussion of regularization and generalization is interesting, but reads more like intuition than validation. Second, the paper omits direct comparison to cascaded low-resolution plus super-resolution approaches, and instead relies on prior work to justify excluding them. That choice is understandable, but the current paper would benefit from making this empirical positioning more explicit. Third, the naming and comparison protocol for the deployment model is confusing: the paper suggests that the deployment version is a distilled/eVAE-equipped T3 model, but Table 7 labels it in a way that can be read as an optimized official Wan baseline. This should be clarified.

Overall, I find the paper technically solid, practically relevant, and supported by meaningful empirical results. The contribution is more on the side of strong engineering and model adaptation than fundamental methodological novelty, but that is still valuable. My concerns are mostly about clarity of positioning and interpretation rather than about the central empirical claim.

---

> ### Author Rebuttal · Authors · 2026-03-31
>
> **Q1: On Regularization and Generalization**
>
> Thank you for your insightful comments.
> 1) Global self-attention has strong expressiveness but lacks local inductive bias (see LocalViT, CvT, CeiT, etc.), making it prone to overfitting long-range spurious correlations under limited data. Local window attention essentially imposes a strong spatial sparsity constraint on the self-attention matrix, acting as a structural regularizer. This effectively reduces the model’s hypothesis space and encourages the network to learn robust local feature representations in shallow layers [1r, 2r]. Relevant evidence will be added in Sec.2.3 of the revised manuscript.
> 2) In Sec.2.3, we analyze the effectiveness of T3 from the perspective of regularization and generalization, and clarify the relationship between structural sparsity and regularization by tightening the generalization bound, supporting the rationality of our design.
> 3) Quantitatively, multi-baseline experiments in Table 6 verify that T3 consistently improves performance and mitigates overfitting across different model sizes and tasks. Table 7 shows that with T3 weight initialization, the full-attention model recovers performance within only 500 iterations, demonstrating preserved learning and generalization ability.
> 4) An in-the-wild 4K Vision World demo is provided in the supplementary materials, validating the effectiveness of T3-Video for native 4K video generation.
>
> [1r] An, Jie, et al. "On Inductive Biases That Enable Generalization in Diffusion Transformers." NeurIPS. 2025.
>
> [2r] Edelman, Benjamin L., et al. "Inductive biases and variable creation in self-attention mechanisms." ICML. 2022.
>
> ---
> **Q2: On Cascaded Low-Resolution + Super-Resolution Approaches**
>
> Thank you for your valuable comments.
> 1) Cascaded methods are generally **not superior** to end-to-end pipelines in efficiency or quality (under similar parameters and compute), limited by repeated feature extraction and accumulated errors. For 4K video generation, high-frequency details produced during super-resolution upsampling often lack semantic coherence, causing temporal flickering and distortion.
> 2) Experiments on UltraGen have confirmed that native generation significantly outperforms cascaded pipelines in video quality and text-image alignment, as it avoids inherent semantic fragmentation in cascaded schemes. Since our method clearly surpasses UltraGen, we did not elaborate further.
> 3) This work focuses on the **native 4K video generation** setting, with the core claim that T3-Video outperforms existing native methods in efficiency and quality for end-to-end native 4K generation. We do not oppose cascaded low-resolution-plus-super-resolution pipelines, which remain beneficial in scenarios with limited GPU memory or cross-team collaboration (e.g., Team A outputs videos for Team B to augment). We will clearly define the scope of this claim in the introduction and conclusion to avoid overgeneralization.
> 4) We will add relevant discussions in the revised related work and Sec.2.4 (Method) to make its empirical positioning more explicit. We will also provide qualitative comparisons of spatial details and temporal stability in the experiments to clearly characterize the performance advantages of T3-Video over cascaded methods.
>
> ---
> **Q3: Naming and Comparison Protocol in Table 7**
>
> Thank you for pointing this out.
> The naming in Table 7 is ambiguous and confusing for readers. The deployed version actually refers to our T3-Video model with step distillation, CFG distillation, and lightweight eVAE, **not** an optimized variant of the official Wan baseline. We have renamed “Wan2.1-T2V-1.3B (Deployment)” to “T3-Video-T2V-1.3B (Deployment Ours)” in Table 7-(e) of the revised manuscript, and added detailed model configurations in Sec.3.2.3.

---

> > ### Author Rebuttal · Reviewer_Q5G9 · 2026-04-03
> >
> > Partially resolved, and I will keep the score.

---

> > > ### Author Response · Authors · 2026-04-03
> > >
> > > Dear Reviewer Q5G9:
> > >
> > > Thank you for recognizing our work. We commit to incorporating the content you suggested in the revised version. Thank you again for your effort in the review and the discussion!
> > >
> > > Best regards!
> > >
> > > Authors of T3-Video

---

### Official Review · Reviewer_ufoz · 2026-03-12

**Soundness:** 3
**Presentation:** 3
**Significance:** 4
**Originality:** 3
**Overall Recommendation:** 4
**Confidence:** 4

**Summary:**

This paper introduces T3-Video, an efficient architecture-adaptation framework for generating 4K videos from pre-trained video diffusion backbones. By incorporating multi-scale shared-window attention and hierarchical optimization, T3-Video outperforms existing methods such as UltraGen in temporal length, quality and efficiency.

**Compliance With Llm Reviewing Policy:**

Affirmed.

**Final Justification:**

The authors have addressed most of my concerns. I encourage the authors to incorporate more quality comparisons in the final version.

**Key Questions For Authors:**

1. UltraGEN reports both classical metrics and VBench results. Can the authors provide similar metrics for a more comprehensive evaluation?
2. The authors should include more discussion and comparisons with other sparse attention methods. If training costs are substantial, could theoretical MACs comparison be provided?
3. After 4K fine-tuning, does the model merely increase sharpness compared to the base model's 720P output? Are there actual improvements in detail and human-perceptible visual quality? Could human evaluation scores or VBench metric comparisons be provided?

**Limitations:**

yes

**Strengths And Weaknesses:**

**Strengths**
1. The paper is clearly written, the method is easy to understand and reproduce.
2. Through extensive experiments, the authors explore effective technical approaches and training strategies for high-resolution video generation.

**Weaknesses**
1. The proposed hybrid window attention essentially implements a fixed sparse attention pattern without specialized design for 4K resolution, making it applicable to various resolutions. Compared to sparse attention optimizations for video diffusion (e.g., NSA or DSV), this method shows no clear advantage in speedup at 480P and 720P resolutions. The authors should include more discussion and comparisons with these approaches. If training costs are substantial, could theoretical MACs comparison and inference latency be provided?
2. Need More Evaluation Metrics: What is the motivation behind 4K-VBench? How do the 4K model outputs perform on VBench directly? What scores would the original base model achieve on VBench-4K?
3. While the authors provide a feasible and relatively efficient solution for 4K video generation that is technically sound and coherent, the actual benefits of this 4K resolution adaptation (e.g., finer details or perceptually higher visual quality) remain unclear. I would like to see visual quality comparisons between the same base model (e.g., Wan-T2V-1.3B baseline at 720P) and T3-Video-T2V-1.3B at 4K after fine-tuning. Does fine-tuning merely increase sharpness compared to the base model's 720P output? Are there actual improvements in detail and human-perceptible visual quality?
4. minor mistake: Line 238-Line239, No Citation; line 278 "some MACs".

[VSA]: Faster video diffusion with trainable sparse attention. \
[DSV]: Exploiting dynamic sparsity to accelerate large-scale video dit training.

---

> ### Author Rebuttal · Authors · 2026-03-31
>
> **Q1: Comparison with Alternatives**
>
> Thanks for your valuable comment.
> 1) In MACs formula (L160), the computational cost of attention rises with resolution. At low resolutions (480p/720p), computation is dominated by linear networks and memory access, leading to lower speedups for T3. This confirms its design focus and practical value for 4K scenarios.
> 2) Trainable VSA/DSV achieve strong performance via heavy fine-tuning, while SVG/STA/RadialAttention emphasize training-free or light-tuning acceleration. We will add relevant discussions in related work and include mechanism comparisons and complexity analyses in the appendix.
> 3) Compared with mainstream methods that offer only 2-4× speedup, our 4.7× speedup remains highly competitive as below:
> |Method|Data Volume|Resolution & Frames|Training GPUs|Training Time/Steps|Speedup & Efficiency|
> |:-:|:-:|:-:|:-:|:-:|:-:|
> |**T3-Video (Ours)**|42K 4K videos|**Native 4K (81 frames)**|64×MI308X|Ultra-fast: **500 steps** for 4K adaptation|43.0× MACs reduction; **21.4× speedup** in 4K;**4.7× speedup** in 720p; 7× faster than UltraGen|
> |**UltraWan/UltraGen**|4K video datasets|4K (Limited **29 frames**)|128×H20|Heavy training/High cost|Baseline inference (Significantly slower than T3)|
> |**STA**|Small/validation datasets|720P (5s videos)|8×H100)|Training-free or lightweight tuning|1.89×-3.53× end-to-end speedup|
> |**VSA**|Large-scale datasets|480P/720P|Up to 64×H200|4,000 steps (approx. 12 hours)|1.7× inference speedup|
> |**Radial Attention**|Small filtered datasets|768P (Long sequence, up to 667 frames)|8×H100|LoRA fine-tuning (139-171 GPU hours)|1.9×-3.7× end-to-end speedup|
> |**DSV**|Large-scale mixed datasets|Various (Up to 520K tokens)|128×H800|Heavy pretraining (Up to 300K steps)|3.02× training throughput improvement|
> |**SVG**|None (Training-free)|720P (80-128 frames)|Training-free|0 steps|2.28×-2.33× end-to-end speedup|
>
> 4) T3-Video focuses on UHD-4K generation and acceleration. We will add corresponding discussions in the “Efficiency analysis” section (L318).
> 5) Training-free and limited-finetuning approaches are generally inferior to trained methods. We therefore further compare our KD deployment version with the strong 3-step variant of VSA on 720p (see **R#acT9-Q2**) to demonstrate the advantages of our method.
>
> ---
> **Q2: Evaluation on 4K-VBench**
>
> Thank you for this excellent question.
> 1) Standard VBench is constrained by the input resolution of its perceptual models, which blurs high-frequency details in 4K videos. Direct evaluation on native 4K also causes out-of-memory errors (L340, Sec.2.6). Accordingly, we propose a high-resolution adapted 4K-VBench with new high-frequency detail metrics (Sec.3.1).
> 2) Table 5 shows official baseline scores on 4K-VBench. Models not adapted for 4K suffer severe performance degradation; for instance, VQA is only 30.01, much lower than T3-Video’s 71.72.
> 3) Following UltraVideo's Table 4, we add a comparison between T3-Video and UltraWan-4K (LoRA training with limited 29 frames due to full-attention computation constraints) on VBench as below (*: Videos downsampled to 1K to avoid OOM). The revised version will include results for the native base model and SoTA UltraGen for a more comprehensive comparison.
>
> |Models|Subject Consistency|Background Consistency|Temporal Flickering|Motion Smoothness|Dynamic Degree|Aesthetic Quality|Imaging Quality|Object Class|Multiple Objects|Human Action|Color|Spatial Relationship|Scene|Appearance Style|Temporal Style|Overall Consistency|
> |---|---|---|---|---|---|---|---|---|---|---|---|---|---|---|---|---|
> |UltraWan-4K (LoRA, 29f)|96.05%|98.02%|98.88%|98.47%*|66.66%*|56.81%|71.61%|50.00%|42.75%|66.66%|100.0%|100.0%|00.00%|19.46%|19.31%|22.88%|
> |T3-Video-4K (FT, 81f)|97.17%|98.10%|98.52%|98.85%*|66.66%*|59.83%|72.18%|66.66%|45.62%|66.66%|100.0%|100.0%|16.66%|19.28%|21.15%|23.61%|
>
> ---
> **Q3: Visual Quality Comparisons**
>
> Thank you for your valuable question.
> 1) T3-Video does not merely perform edge sharpening; it generates physically plausible high-frequency textures via multi-scale attention. It outperforms baselines in both spatiotemporal detail metrics and VQA (Table 5). 4K quantitative results (Table 5) are also significantly better than 720p results in Table 7, verifying the effectiveness of 4K fine-tuning.
> 2) Human preference studies in Table 8 show that T3-Video is strongly preferred for its rich details, with visual improvements aligned with human perception.
> 3) We will add zoomed local comparisons between 720p and 4K results in the appendix to clearly demonstrate the quality of added details.
>
> ---
> **Q4: Minor Mistakes**
>
> Thank you for pointing out these errors.
> 1) We have added the corresponding domain attention reference [NAT] for the missing citation.
> 2) The typo in the Table 1 caption has been corrected: "some" → "same".
>
> [NAT] Hassani, Ali, et al. "Neighborhood attention transformer." CVPR. 2023.

---

> > ### Author Rebuttal · Reviewer_ufoz · 2026-04-03
> >
> > Most of my concerns are addressed.

---

> > > ### Author Response · Authors · 2026-04-03
> > >
> > > Dear Reviewer ufoz:
> > >
> > > Thank you for recognizing our work. We commit to incorporating the content you suggested in the revised version. Thank you again for your effort in the review and the discussion!
> > >
> > > Best regards!
> > >
> > > Authors of T3-Video

---

### Official Review · Reviewer_acT9 · 2026-03-13

**Soundness:** 2
**Presentation:** 2
**Significance:** 2
**Originality:** 2
**Overall Recommendation:** 4
**Confidence:** 4

**Summary:**

This paper proposes T3-Video (Transform Trained Transformer), a method for accelerating native 4K video generation efficiently. The key idea is to transform global full-attention into a multi-scale shared window attention mechanism, which preserves compatibility with pretrained weights while reducing attention complexity from quadratic to approximately linear with respect to token count. The approach enables efficient fine-tuning of pretrained video diffusion models and significantly reduces compute requirements. Experiments on 4K video generation benchmarks show improvements in both efficiency and generation quality compared with existing methods.

**Compliance With Llm Reviewing Policy:**

Affirmed.

**Final Justification:**

They address most of my concerns successfully.

**Key Questions For Authors:**

N/A

**Strengths And Weaknesses:**

Strengths:

1. The paper focus on the quadratic complexity attention in 4K video generation. This problem is known to cause great inefficiency.
2. The approach can integrate with distillation, LoRA, and efficient VAE components.
3. T3 can degenerate to full attention or a linear layer, indicating strong flexibility and expressiveness.

Weaknesses:

1. Windowed attention and multi-scale attention patterns are well studied, and the conceptual difference from existing approaches is not fully clarified. How does the proposed attention transformation differ fundamentally from existing window or sparse attention mechanisms?
2. Some alternative attention or sparse mechanisms are not thoroughly evaluated. For example, SVG / STA / RadialAttention
3. What are the trade-offs between window size, number of scales, and generation quality?
4. The method’s practical speedup only approaches its theoretical gains at 4K resolution. Even at lower resolutions such as 1080p the actual speedup is 4x lower than theoretical speedup.

---

> ### Author Rebuttal · Authors · 2026-03-31
>
> **Q1: Difference from existing window or sparse attention mechanisms**
>
> Thanks for the comment.
> 1) **Mechanism**: Existing window attention performs single-scale local modeling and requires shifted windows or stacked layers for global interaction; a single layer cannot model global semantics, leading to inferior performance (Table 7-Local Type ablation). T3 models both local details and global semantics in one layer (L126). Sparse attention relies on spatial-temporal redundancy and is a parallel, compatible approach with T3. Since T3's inner-window attention is unchanged, SVG/STA/RadialAttention can be reused directly (L84).
> 2) **Architecture unification**: Existing sparse/window attention uses customized designs, causing hardware compatibility issues and poor inheritance of pre-trained knowledge. T3 degenerates to full attention (window=resolution) or linear layers (window=1) (L255 in Sec.2.3), naturally inheriting full-attention modeling ability and pre-trained knowledge. Window constraints also act as regularization to reduce overfitting.
> 3) **Training efficiency**: T3’s parameter sharing provides inductive bias consistent with full attention, enabling fast high-resolution adaptation with few iterations, outperforming sparse methods requiring heavy training. Pure 4K fine-tuning achieves strong results in only 2,000 steps (L310 & Fig. 6); the progressive scheme needs just 500 steps (L303). By comparison, SoTA VSA requires 64 H200 GPUs (our MI308X is H20-class) for 4,000 steps to generate 448×832×61f and cannot natively output 2176×3840×81f.
> 4) **Performance**: T3 enables native 4K video generation with 21×actual speedup (43×theoretical). Comparable methods only support 720P with 2-4×speedup, far below T3.
> 5) **Implementation**: T3 only modifies the attention forward pass and can be implemented via one-line code replacement, making it concise and elegant.
> 6) **Revision**: The updated Sec.2.2 will expand related work discussions to clarify innovations and core differences.
>
> ---
> **Q2: Comparison with alternative attention or sparse mechanisms**
>
> 1) SVG relies on online analysis to differentiate attention heads; STA focuses on sliding block kernel optimization; RadialAttention uses energy decay masks. These methods target training-free or lightweight fine-tuning for acceleration. Trainable VSA/DSV require heavy computational fine-tuning for gains.
> 2) T3 is an architecture-level transformation that enables efficient 4K fine-tuning with semantic consistency without specialized hardware kernel optimization. It is compatible with mainstream native attention optimizers as inner-window attention logic is unmodified.
> 3) T3-Video explores native 4K generation with supporting optimizations (new T3, CFG/Step KD, efficient VAE engineering). Above methods cannot achieve 4K generation and thus are not directly compared. We will add discussions in related work and include mechanism comparisons and complexity analyses in the appendix.
> 4) Training-free and limited fine-tuning methods are generally inferior to fully trained solutions. We compared T3's KD deployment version against the strong VSA 3-step version on 720P, verifying T3’s advantages.
> |Model|VQA|VTC|DoG|BM|RA|TDS|TEP|Speedup|
> |:-:|:-:|:-:|:-:|:-:|:-:|:-:|:-:|:-:|
> |Wan-1.3B (Official)|70.56|0.91|0.42|0.938|0.71|0.87|0.51|1x|
> |VSA|68.92|0.89|0.38|0.935|0.71|0.88|0.50|50.9x|
> |Ours|69.37|0.90|0.40|0.948|0.72|0.89|0.49|111.3x|
>
> ---
> **Q3: Trade‑offs between window size, number of scales, and generation quality**
>
> Thanks for your valuable advice.
> 1) Larger window sizes and more scales effectively expand the global receptive field and improve spatiotemporal coherence, but push complexity toward quadratic growth; excessive scales also increase memory and computation overhead.
> 2) Default scale number and partitioning ratios in this paper are optimized under memory constraints, balancing generation quality and hardware practicality (L199).
> 3) We will add experiments with varying window sizes (WS) and Scale (S) to extend Table 7 (using the nearest valid 720P resolution for divisibility). Results (see **R#Dr7s-Q1**) show that larger window sizes yield slight quality gains but increase attention MACs.
> 4) This inspires us to explore the relationship between resolution and window size/scales in future work.
>
> ---
> **Q4: Speedup at 1080p**
>
> 1) From MACs formula (L160), the attention computation ratio rises with resolution. At 1080p and below, latency is dominated by linear networks and memory access.
> 2) Theoretical speedup only accounts for attention MAC reduction (only attention part in Table 1). Actual end-to-end DiT latency (Table 2) is affected by hardware, operator scheduling and non-attention modules, causing the gap.
> 3) Compared with existing methods with 2-4×speedup, our 4.7×speedup remains highly competitive (see **R#ufoz-Q1**).
> 4) This work focuses on UHD-4K generation and acceleration. We will add relevant analysis in the efficiency analysis section (L318).

---

> > ### Author Rebuttal · Reviewer_acT9 · 2026-04-03
> >
> > I would still like to see a more detailed discussion of sparse attention methods in the related work section, including both training-free and training-based approaches. I also encourage the authors to include at least one experimental comparison with such methods. Other than this, I think the concerns have been addressed reasonably well, so I have increased my rating. I hope the authors can incorporate these improvements in the camera-ready version.

---

> > > ### Author Response · Authors · 2026-04-03
> > >
> > > Dear Reviewer acT9:
> > >
> > > Thank you for recognizing our work. We commit to incorporating the content you suggested in the revised version. Thank you again for your effort in the review and the discussion!
> > >
> > > Best regards!
> > >
> > > Authors of T3-Video

---

### Decision · Program_Chairs · 2026-04-30

**Decision:**

Accept (regular)

**Comment:**

This submission presents T3-Video, a practical transformer-retrofitting framework for native 4K video generation that transforms pretrained full-attention models into multi-scale shared-window attention while preserving compatibility with pretrained weights. The reviewers were generally positive, with three Weak Accept and one Accept. After rebuttal, most concerns were at least partially resolved, especially regarding the difference from existing sparse/window attention and evaluation on 4K benchmarks. Overall, the paper is technically sound and practically important, and the rebuttal addressed most major issues, so an accept recommendation is suggested. However, the final version should more clearly state its empirical scope and strengthen related-work/comparison discussion.